

# Recognition of sports and daily activities through deep learning and convolutional block attention

Sakorn Mekruksavanich[1], Wikanda Phaphan[2], Narit Hnoohom[3] and Anuchit Jitpattanakul[4,5]

[1] Department of Computer Engineering, School of Information and Communication Technology, University of Phayao, Phayao, Thailand
[2] Department of Applied Statistics, Faculty of Applied Science, King Mongkut's University of Technology North Bangkok, Bangkok Thailand
[3] Department of Computer Engineering, Faculty of Engineering, Mahidol University, Nakhon Pathom, Thailand
[4] Department of Mathematics, Faculty of Applied Science, King Mongkut's University of Technology North Bangkok, Bangkok, Thailand
[5] Intelligent and Nonlinear Dynamic Innovations Research Center, Science and Technology Research Institute, King Mongkut's University of Technology North Bangkok, Bangkok, Thailand

Corresponding author
Anuchit Jitpattanakul,
anuchit.j@sci.kmutnb.ac.th

## ABSTRACT

Portable devices like accelerometers and physiological trackers capture movement and biometric data relevant to sports. This study uses data from wearable sensors to investigate deep learning techniques for recognizing human behaviors associated with sports and fitness. The proposed CNN-BiGRU-CBAM model, a unique hybrid architecture, combines convolutional neural networks (CNNs), bidirectional gated recurrent unit networks (BiGRUs), and convolutional block attention modules (CBAMs) for accurate activity recognition. CNN layers extract spatial patterns, BiGRU captures temporal context, and CBAM focuses on informative BiGRU features, enabling precise activity pattern identification. The novelty lies in seamlessly integrating these components to learn spatial and temporal relationships, prioritizing significant features for activity detection. The model and baseline deep learning models were trained on the UCI-DSA dataset, evaluating with 5-fold cross-validation, including multi-class classification accuracy, precision, recall, and F1-score. The CNN-BiGRU-CBAM model outperformed baseline models like CNN, LSTM, BiLSTM, GRU, and BiGRU, achieving state-of-the-art results with 99.10% accuracy and F1-score across all activity classes. This breakthrough enables accurate identification of sports and everyday activities using simplified wearables and advanced deep learning techniques, facilitating athlete monitoring, technique feedback, and injury risk detection. The proposed model's design and thorough evaluation significantly advance human activity recognition for sports and fitness.

# INTRODUCTION

The study of human activity recognition (HAR) is gaining momentum across various fields, focusing on automatically identifying bodily movements. There are two main approaches to performing HAR: the computer vision method and the sensor-based method (*Diraco et al., 2023*). In the computer vision approach, video material is closely examined to identify human behaviors, showing optimal performance in controlled laboratory environments. However, practical scenarios pose challenges, with factors like ambient noise, varying illumination, and contrast discrepancies potentially hindering the effectiveness of computer vision and causing malfunctions. On the other hand, the sensor-based method relies on data from wearable devices and ambient sensors to recognize and classify activities. This approach proves to be more robust for HAR in unpredictable, real-time situations with potential visual disruptions. As researchers enhance the capabilities of HAR, they persist in exploring both methodologies to comprehend their respective advantages and limitations (*Zhang et al., 2022*).

HAR has emerged as an essential technology in numerous sports and fitness applications (*Pajak et al., 2022a*; *Hussain et al., 2022*; *Steels et al., 2020*; *Pajak et al., 2022b*). By automating the identification of movements and workouts using sensor data, it becomes possible to gauge, document, and assess the level of physical activity carried out by athletes. This facilitates functionalities like automated workout monitoring, form evaluation, repetition counting, technique rating, and more. However, accurately recognizing and categorizing physical movements in sports poses unique and intricate challenges. These movements are often elaborate and intense, involving multiple overlapping actions that engage the entire body. Activities within the same category may display subtle differences influenced by biomechanics, anatomy, injury history, and individual skill levels. Moreover, several exercises incorporate transitional movements or non-standardized sequencing between repetitions. These attributes complicate various actions by human detection techniques designed for diverse purposes. Efficient algorithms for sports applications must be capable of identifying a diverse array of dynamic, detailed movements while also adjusting to individual variations.

In the past ten years, machine learning methodologies have unveiled resilient and intricate capabilities for recognizing activities in sports and fitness applications. *Bian et al. (2019)* introduced a system for categorizing weight training activities, utilizing an accelerometer and three force sensors. K-nearest neighbor, decision tree, and random forest models underwent training and testing, achieving high accuracies exceeding 95%. The research (*Zebin, Scully & Ozanyan, 2017*) compared supervised classification algorithms for HAR. Diverse statistical time and frequency domain features were extracted from inertial sensor data. Another aspect of the study (*Nurhanim et al., 2017*) also focused on classifying signals from smartphone inertial sensors, employing the support vector machine algorithm in both time and frequency domains. Nevertheless, conventional machine learning relied on manually crafted features derived from sensor data, demanding significant expertise in the specific field and often demonstrating poor generalization. Recent approaches based

on deep neural networks can autonomously learn latent feature representations directly from raw sensor streams.

Recent breakthroughs in advanced learning techniques have propelled cutting-edge achievements in addressing intricate challenges related to pattern recognition across diverse domains (*Ascioglu & Senol, 2023*; *Wu, Zhu & Wan, 2023*; *Gou et al., 2023*). Models such as convolutional neural networks (CNNs), recurrent neural networks (RNNs), and attention networks have effectively handled high-dimensional time series data. These temporal models can implicitly grasp and assess intricate concepts like motion, sequencing, transitions, intensity, duration, and coordination. Simultaneously, the fusion of data from various sensors through multi-modal sensor integration, including accelerometers, gyroscopes, magnetometers, barometers, heart rate monitors, and more, facilitates capturing biomechanical and physiological aspects. The synergy of deep learning and wearable sensors enables inconspicuous monitoring of subtle motion patterns, fatigue levels, workout quality, improper form, and factors contributing to the risk of injury. The most recent techniques lay the groundwork for a new era in sports tracking technology, emphasizing optimization, prevention, and personalization for casual and elite athletes.

Despite advancements in HAR through deep learning techniques, further research is necessary to develop robust and efficient models tailored for identifying a broad spectrum of sports and everyday activities utilizing data from wearable sensors. Present studies often concentrate on limited tasks or rely on complex sensor setups that could be more practical for real-world applications. Additionally, publicly available datasets encompassing diverse sport-related activities are scarce, impeding the creation and evaluation of models proficient in recognizing various sports and fitness movements.

We propose a novel hybrid deep learning framework called CNN-BiGRU-CBAM to address this research gap. This model integrates CNNs, bidirectional gated recurrent unit networks (BiGRU), and convolutional block attention modules (CBAM) to classify sports and daily activities accurately using wearable sensor data. The primary objectives of this study are as follows:

- Develop a sophisticated deep learning model that accurately identifies various sports and everyday activities utilizing efficient wearable sensors.
- Evaluate the performance of the proposed model using benchmark datasets containing diverse sports and daily activities and compare its efficacy with other state-of-the-art deep learning models.
- Investigate how sensor placement on the body influences activity detection performance to achieve optimal outcomes.

We aim to advance HAR in sports and fitness contexts, facilitating the development of more precise and practical activity monitoring systems beneficial for athletes and individuals in their training and daily routines.

This article delves into customized deep-learning methodologies to classify sports and daily activities accurately. The focus is on utilizing time-series data collected from wearable sensors. A hybrid deep neural network is proposed in this work for HAR using wearable sensors. The architectural design of this model leverages the complementary strengths of

CNNs for spatial feature extraction, BiGRUs for temporal context modeling, and CBAM for attention-based emphasis on informative features. In summary, this study's contribution can be encapsulated as follows:

- This study aims to develop a novel deep learning architecture known as CNN-BiGRU-CBAM, explicitly tailored to precisely identify sports and daily activities using data from wearable sensors. This model combines CNNs, BiGRUs, and CBAM to effectively extract spatial features, capture temporal relationships, and prioritize relevant areas in sensor data. The proposed framework represents HAR advancement, particularly in sports and fitness.

- The CNN-BiGRU-CBAM model is subjected to a rigorous evaluation using a benchmark dataset encompassing a wide range of sports and everyday activities. The performance assessment, which includes multi-class classification metrics such as accuracy, precision, recall, and F1-score, reveals the model's exceptional capabilities. The results demonstrate that the proposed model outperforms existing advanced deep learning models, achieving an average accuracy and F1-score of 99.10% across all activity categories, a testament to its superior performance.

- The study delves into the crucial aspect of sensor placement on the body, aiming to determine the optimal positioning for accurate activity recognition performance. This research yields invaluable insights into the most effective sensor locations for detecting sports and daily activities. By analyzing model performance using sensor data from various body positions, including the torso, right arm, left arm, right leg, and left leg, these findings provide practical guidance for the development of more efficient and user-friendly wearable sensor systems for HAR applications.

- A comprehensive ablation study is conducted to assess the specific effects of critical components in the CNN-BiGRU-CBAM model. This research elucidates the significance of each element—CNN, BiGRU, and CBAM modules—by systematically removing or replacing them. It highlights their contributions to the overall model performance, validating the design choices in the proposed architecture and guiding future research in crafting efficient HAR models for sports and fitness applications.

The article is organized as follows: Section 'Related Works' examines the current landscape of applying deep learning to recognize sports and daily activities, highlighting challenges and constraints. Section 'The Proposed Methodology' explains the proposed approach and the structure of the deep learning model designed for recognizing activities; section 'Experiments and Research Findings' details the experiment setup, benchmark datasets, metrics for evaluation, and the presentation of results. Additionally, a thorough analysis of the model's exceptional performance is included. Section 'Discussion' of the research delves into various significant and captivating points derived from the findings. Finally, 'Conclusion and Future Works' summarizes the article's contributions and suggests future research methods.

## RELATED WORKS

Recognition of human activity is attracting substantial attention in research, driven by its potential applications in healthcare, sports, and the monitoring of general human movement. Numerous approaches based on machine learning have been introduced to tackle HAR in the past few years. This section reviewed works on recognizing sports and fitness activities and explored deep-learning frameworks designed for HAR.

### Traditional machine learning approaches for HAR

Early HAR research primarily utilized traditional machine-learning techniques. For instance, *Siirtola et al. (2011)* applied linear discriminant analysis (LDA) on acceleration data sampled at 5 Hz to classify swimming activities and count strokes. Similarly, *Ponce, Miralles-Pechuán & Martınez-Villaseñor (2016)* employed an artificial intelligence network to categorize daily actions such as sitting, standing, reclining, lying on one side, and playing basketball. These studies showcased the potential of machine learning in HAR but often concentrated on a restricted set of activities and necessitated manual feature engineering.

### Deep learning approaches for HAR

Recent progress in deep learning has transformed the field of HAR by facilitating automated feature extraction and enhancing overall performance. *Hammerla, Halloran & Plötz (2016)* extensively explored deep neural networks, CNNs, and RNNs, performing practical assessments on three openly available datasets. The results of their study indicated that the most effective method for scrutinizing jogging and walking activities was the CNN-based approach. Another inquiry by *Coelho et al. (2019)* introduced a HAR system grounded in CNNs for categorizing six distinct activities: running, walking, ascending and descending stairs, standing, and sitting. This approach exhibited noteworthy levels of accuracy and precision, reaching 94.89% and 95.78%, respectively. Furthermore, *Lee, Yoon & Cho (2017)* proposed a HAR technique utilizing a one-dimensional CNN and three-axis accelerometer data from mobile devices.

Although deep learning methodologies have demonstrated encouraging outcomes, they frequently concentrate on restricted activities. They may need help effectively capture the intricate temporal relationships in sensor data. Furthermore, the need for more extensive and varied publicly available datasets for sports and fitness-related activities impedes advancing and assessing HAR models within these realms.

Several deep learning architectures have been specifically tailored and assessed using the UCI-DSA dataset, which was also employed in our study. GoogLeNet, proposed by *Szegedy et al. (2015)*, introduces the inception module within a deep CNN structure to enhance performance by capturing multiple-scale features. ResNeXt, introduced by *Xie et al. (2017)*, extends the ResNet framework by integrating the inception module with a residual structure, enabling the model to learn more intricate and varied features. Multi-STMT, proposed by *Zhang & Xu (2024)*, is a multi-level network that combines a CNN module, a BiGRU module, and an attention mechanism to capture spatial and temporal dependencies within sensor data. *Tuncer et al. (2020)* presented an automated method for recognizing daily sports activities and gender based on a novel multikernel local diamond pattern using

sensor signals, illustrating the immense potential of deep learning methods for specific HAR tasks.

Additionally, _Khatun et al. (2022)_ introduced a deep CNN-LSTM model with self-attention for HAR using wearable sensors, demonstrating the efficacy of integrating various deep learning architectures. Meanwhile, _Mim et al. (2023)_ proposed GRU-INC, an approach based on inception and attention mechanisms utilizing GRU for HAR, further underscoring the significance of attention mechanisms in capturing pertinent features.

Despite their promising outcomes, these deep learning methodologies often concentrate on a restricted range of activities and may not effectively capture the intricate temporal relationships within sensor data. Furthermore, the absence of comprehensive and diverse public datasets for sports and fitness activities poses challenges for the development and assessment of HAR models in these domains.

### Sport-specific HAR datasets and challenges

Publicly accessible datasets for HAR, especially those containing sensor data from sports and fitness activities, are relatively uncommon. Frequently, existing datasets like UCI-DSA (_Barshan & Altun, 2013_) and the study conducted by _Trost, Zheng & Wong (2014)_ consolidate various sports into a singular dataset, thereby diminishing the granularity of specific sports actions by categorizing them under broad target classes. The UTD Multimodal Human Action Dataset (_Chen, Jafari & Kehtarnavaz, 2015_) incorporates sports activities; however, it is constrained to a solitary action per sport, rendering it an inadequate representation of the comprehensive nature of each sport.

In recent times, the dataset for basketball activities introduced by _Hoelzemann et al. (2023)_ has gained prominence due to its diverse range of classes, a sizeable number of participants involved in the study, and its comprehensive inclusion of the intrinsic characteristics of basketball that vary based on the ruleset. Nonetheless, a pressing requirement remains for additional publicly accessible datasets focused explicitly on sports activities, which can effectively capture the inherent variability and complexity present within these domains, utilizing inertial sensor data as the primary source.

### Addressing the gaps and proposed framework

The framework proposed in this study, CNN-BiGRU-CBAM, addresses the constraints of prevailing HAR methodologies and datasets. By synergizing CNNs for spatial feature extraction, BiGRU for temporal context modeling, and CBAM for attention-based feature refinement, our model effectively captures both spatial and temporal interdependencies present in sensor data.

Furthermore, we meticulously evaluate our model's performance on the UCI-DSA dataset, which covers a wide range of sports and daily activities. This rigorous assessment highlights the robustness and generalizability of our approach, demonstrating its superiority over state-of-the-art techniques.

## THE PROPOSED METHODOLOGY

The focus of this study was the recognition of sport and fitness activities (SDAR) using deep learning methods to abstract features from raw sensor data. Illustrated in Fig. 1, the

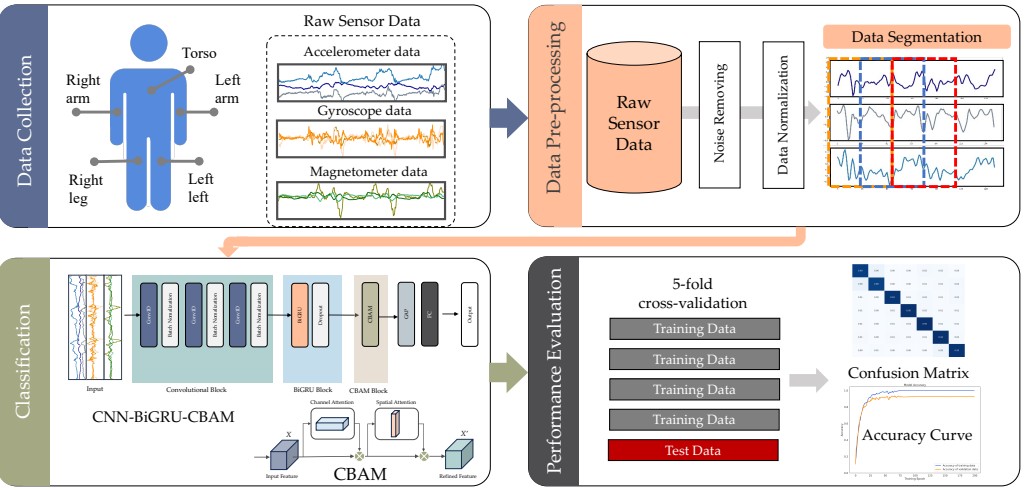

**Figure 1** SDAR workflow used in this study.

SDAR framework under examination comprises four essential stages: data collection, data pre-processing, classification, and performance evaluation.

## Data collection

We possess only one public dataset that contains activities with non-repetitive and intricate motion states, specifically the Daily and Sports Activity dataset (UCI-DSA) *Barshan & Yüksek (2014)*. This dataset includes sensor data that captures 19 distinct physical actions undertaken by eight participants (four female, four male, ages 20–30). Each activity was executed for 5 min by each participant, generating 5 min of signal data per participant for each movement. The signals were recorded at a sampling frequency of 25 Hz and segmented into 5-second intervals, resulting in 480 segments for each action (60 segments per participant).

The UCI-DSA encompasses 19 movements, namely: seating, standing, lying (on the back and right side), ascending and descending stairs, standing and moving in an elevator, strolling (in a parking lot and on a 4 km/h treadmill-flat and 15 deg incline), running on an 8 km/h treadmill, stepping, cross training, cycling (horizontal and vertical), rowing, jumping, and playing basketball.

For data collection, inertial measurement units (IMUs) were utilized, and each participant affixed these units to five different body parts. Each IMU was equipped with three sensors for acceleration, three for rotational velocity, and three for the magnetic field, facilitating the capture of intricate motion details. Throughout the data collection process, participants attached one IMU to their torso, right and left arm, right and left leg, completing a comprehensive sensor configuration covering the entire body (*Barshan & Yüksek, 2014*). This configuration allowed the recording of diverse, multi-modal signals,

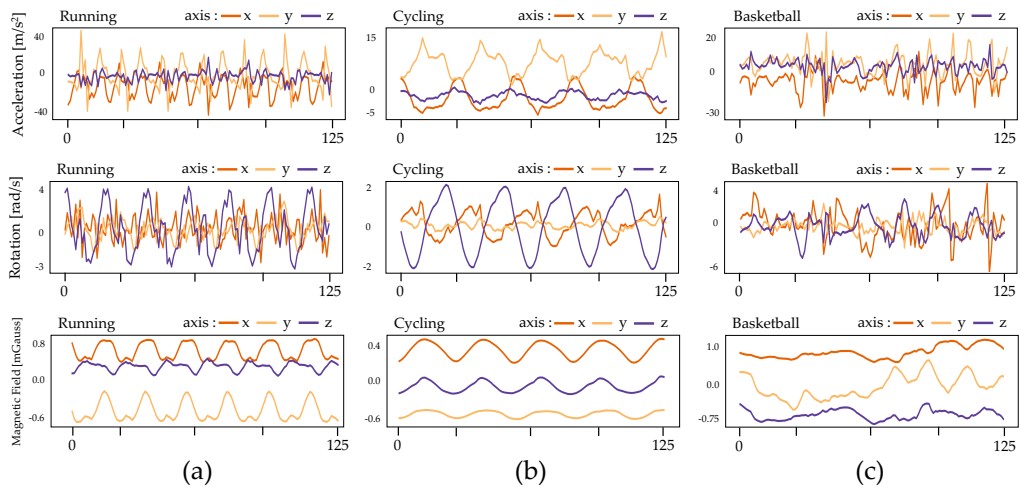

**Figure 2** Illustration of IMU sensor data of three types of sports: (A) SP-SMS, (B) WP-CMS, and (C) NP-CMS.

encompassing movements from all major body parts. Figure 2 displays some examples of the collected IMU data.

Figure 2 portrays examples of sensor data from inertial measurement units (IMUs) for three sports activities. In the case of SP-SMS activity (such as running), the acceleration plot exhibits repetitive patterns along the $x$, $y$, and $z$ axes, representing the periodic limb motions inherent in running. The rotational velocity also displays periodic patterns corresponding to the rotational motion of the limbs/torso, while the magnetic field remains relatively stable during this repetitive activity.

For WP-CMS activity (like cycling), the acceleration plot shows less distinct periodic patterns than running due to the more intricate multi-state motions involved. Similarly, rotational velocity and magnetic fields exhibit reduced periodicity during complex basketball maneuvers.

In NP-CMS activity (for example, basketball), the acceleration, rotational velocity, and magnetic fields lack periodic or repetitive patterns. This absence indicates more random and non-periodic motions during the activity.

## Data pre-processing

Processing data from wearable sensors often involves preparatory steps before constructing a model, aiming to convert raw signals into indicative features (*Zheng, Wang & Ordieres-Meré, 2018*). In this study, a systematic pre-processing pipeline for structured data was applied before the development of deep networks. This pipeline included denoising, normalization, and segmentation of the data.

### Data denoising

The raw data acquired from sensors carried measurement noise from system limitations or unexpected movements during the experiments. This noisy signal interferes with the

valuable information embedded in the data. Consequently, it was imperative to mitigate the impact of noise to extract meaningful information for subsequent processing. Various filtering methods are commonly employed to handle this issue, including the mean filter, low-pass filter, Wavelet filter, and Gaussian filter (*Rong et al., 2007*; *Mostayed et al., 2008*). In this study, we opted for an average smoothing filter to denoise the signal, applying the filter across all three dimensions of the accelerometer and gyroscope sensors.

### Data normalization

Subsequently, the raw data from sensors undergo normalization, ensuring their values fall within the range of 0 to 1, as shown in Eq. (1). This step addresses challenges in model learning by harmonizing the data values to a uniform range, facilitating quicker convergence during gradient descents.

$$X_i^{norm} = \frac{X_i - x_i^{\min}}{x_i^{\max} - x_i^{\min}}, i = 1, 2, 3, \ldots \tag{1}$$

where $X_i^{\mathrm{norm}}$ represented the normalized data, $n$ represented the number of channels, $x_i^{\max}$ and $x_i^{\min}$ are the maximum and minimum values of the $i$th channel, respectively.

### Data segmentation

As wearable sensors produce a large amount of signal data, it is impractical to simultaneously input all of this data into the HAR model. Therefore, the data must be divided into segments using a sliding window approach before being fed into the model. Splitting data into windows is a widely employed technique in HAR to separate data for analysis. It is commonly used to detect recurring activities such as walking, jogging, and inactive, including standing, sitting, and lying down (*Banos et al., 2014*). The continuous sensor signals are split into fixed-length windows. Consecutive windows overlap partially to increase the number of available training examples and avoid missing transitions between activities. Figure 3 demonstrates the windowing process.

The segmented sample data, obtained using a sliding window of size $N$, has a dimension of $K \times N$. The given sample is represented by the symbol $W_t$.

$$W_t = [a_t^1 a_t^2 \ldots a_t^K] \in \mathbb{R}^{K \times N} \tag{2}$$

The column vector $a_t^k = (a_t^1, a_t^2, \ldots, a_{t_N}^k)^T$ represents the signal data of sensor $k$ at window time $t$. T denotes the transpose operator, $K$ represents the number of sensors, and $N$ indicates the length of the sliding window. To exploit the relationships between windows and facilitate the training process, the window data is divided into sequences of windows.

$$S = \{(W_1, y_1), (W_2, y_2), \ldots, (W_T, y_T)\} \tag{3}$$

$T$ is the length of the window series, and $y_t$ represents the activity label of the associated window $W$. The label of a window that contains several activity classes will be determined by selecting the most often occurring sample activity.

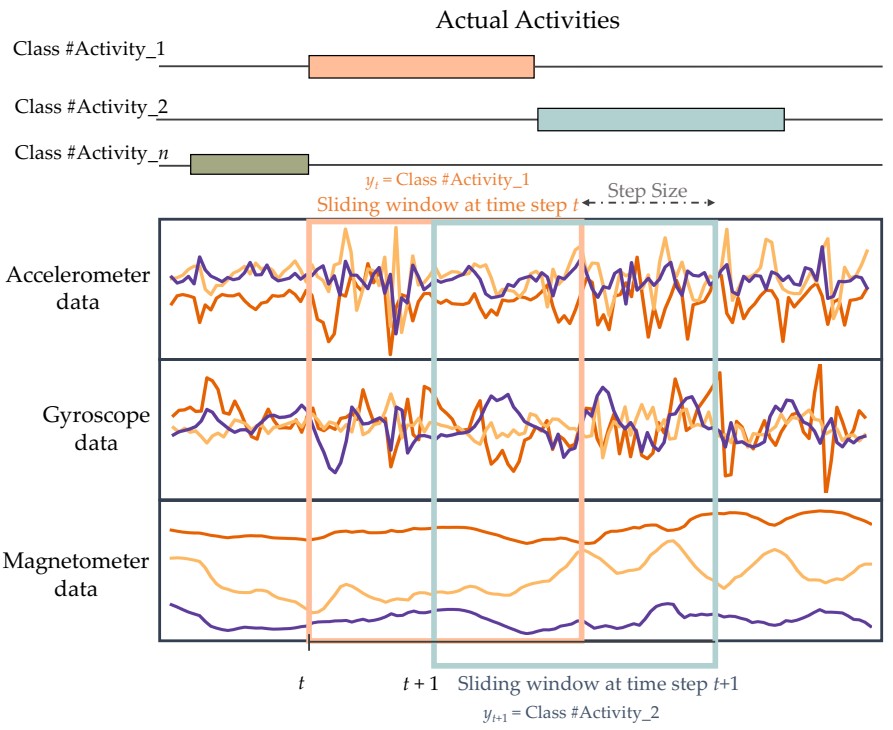

**Figure 3** Illustration of fixed-length sliding window technique used in this work.

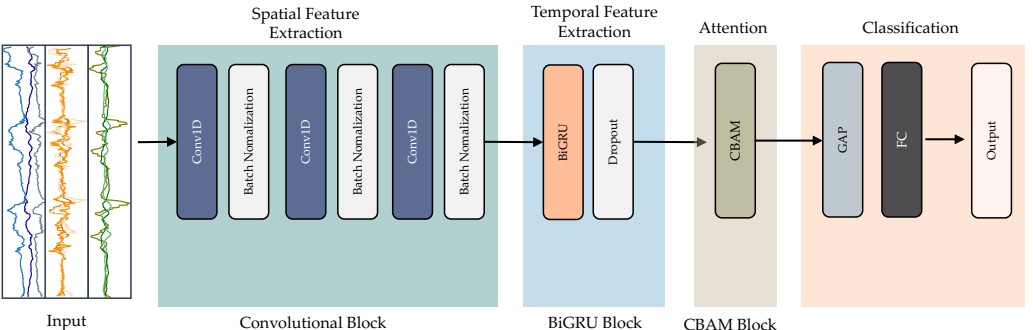

**Figure 4** Detailed and architecture of the proposed CNN-BiGRU-CBAM model.

## The proposed CNN-BiGRU-CBAM model

The suggested model utilizes an end-to-end deep learning architecture. This hybrid design incorporates three key elements: convolution, residual bidirectional GRU, and CBAM blocks. The convolution extracts features from the input data. The bidirectional GRU retains information from both directions in time. Residual connections allow information

to shortcut across layers. The attention mechanism focuses the model on relevant features. Figure 4 illustrates the entire structure of the proposed integrated deep learning model.

The first element, the convolutional block, extracts spatial characteristics from the pre-processed data. Modifying the convolution kernel's step size shortens the time series length, quickening recognition time. The next component employs a BiGRU network to capture temporal patterns in the data after convolution. Integrating BiGRU capacities strengthens the model's ability to incorporate long-term dependencies in the time series. This fusion enhances the model's capacity to comprehend intricate temporal features, improving accuracy.

We then apply an attention mechanism called CBAM to refine the final recognition characteristics. This module calculates weights for the information generated by the BiGRU, enabling the model to focus on the most informative input data. Highlighting the most relevant elements reinforces the model's ability to discriminate between activities, increasing recognition accuracy.

Finally, a fully connected layer and SoftMax function categorize the behavior details. The output prediction specifies the activity performed. In the following sections, we elaborate on each component, delineating their roles within the proposed model.

### Convolution block

CNNs rely on a standard set of components. They often apply supervised learning methods. Typically, CNN neurons connect to every neuron in the next layers. An activation function transforms the neuron's inputs into outputs. Two key factors impact the activation function's efficacy. One is sparsity—the fraction of zero activations. Another is ensuring sufficient gradient flow as the network depth increases. CNNs frequently use pooling to reduce dimensionality. Two common techniques are taking the maximum (max-pooling) or average (average-pooling) value over input regions. Max-pooling and average-pooling simplify the representations by reducing the number of parameters within activation maps. At the same time, they preserve the most salient information.

This work involves employing convolutional blocks (ConvB) for identifying essential characteristics in raw sensor data. The diagram in Fig. 4 demonstrates that ConvB comprises six layers of 1D-convolutional (Conv1D) and batch normalization (BN). The structure is made up of three layers of Conv1D-BN. Conv1D employs multiple trainable convolutional kernels to capture diverse attributes, with each kernel producing a distinct feature map. To optimize the training process's efficiency and speed, we opted to incorporate the BN layer.

### BiGRU block

At the core of our proposed CNN-BiGRU-CBAM model lies the BiGRU. This crucial component captures temporal correlations and context from the features extracted by the convolutional layers. Unlike conventional RNNs, BiGRU excels in processing sequential input by maintaining an internal state and employing gating mechanisms to control information flow.

In a standard GRU, the hidden state $h_t$ at time $t$ isdetermined by combining the outputs of the update gate $z_t$, reset gate $r_t$, current input $x_t$, and the previously hidden state $h_{t-1}$. The update gate $z_t$ determines the amount of information to retain from the previous hidden

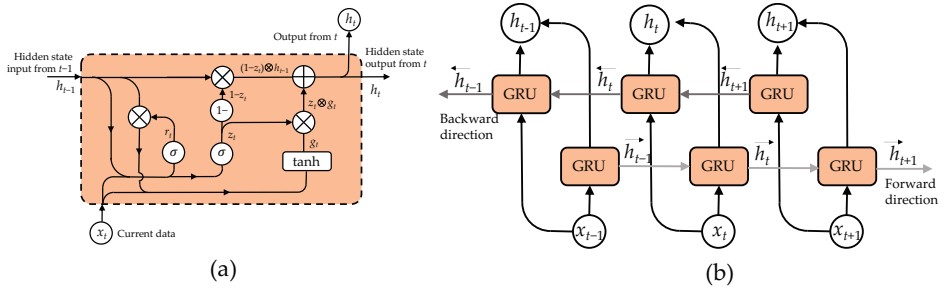

**Figure 5** **Structure of bidirectional GRU (BiGRU): (A) GRU cell and (B) unroll BiGRU.**

state, while the reset gate $r_t$ determines the degree to which the previous hidden state should be ignored. Figure 5A illustrates the structure of a GRU cell and the mathematical equations managing its operation as follows:

$$z_t = \sigma(W_z x_t \oplus U_z h_{t-1}) \tag{4}$$

$$r_t = \sigma(W_r x_t \oplus U_r h_{t-1}) \tag{5}$$

$$g_t = \tanh(W_g x_t \oplus U_g(r_t \otimes h_{t-1})) \tag{6}$$

$$h_t = ((1 - z_t) \otimes h_{t-1}) \oplus (z_t \otimes g_t) \tag{7}$$

Here, $\sigma$ represents a sigmoid function, $\oplus$ denotes an element-wise addition, and $\otimes$ denotes an element-wise multiplication.

To augment the capacity of the GRU in apprehending historical and forthcoming context, we integrate a BiGRU into our framework. Illustrated in Fig. 5B, a BiGRU comprises two distinct GRU strata: a forward GRU, which handles the input sequence from the initial timestep to the concluding one, and a backward GRU, which inversely manages the sequence. Subsequently, the outputs of both GRU layers are combined at each timestep to generate the ultimate BiGRU output, encompassing insights from both antecedent and subsequent contexts.

Within our CNN-BiGRU-CBAM architecture, the BiGRU segment receives the spatial characteristics derived from the convolutional strata as its input. Independently, the forward and backward GRU strata analyze these characteristics, grasping the temporal connections and context in opposing directions. By amalgamating the results of both GRU layers, the BiGRU component furnishes a more exhaustive comprehension of the temporal correlations among the extracted features. Consequently, this enhancement empowers the model with improved capabilities to discern and categorize diverse sports and daily endeavors.

Incorporating a BiGRU component in our model presents numerous benefits compared to a single-direction GRU. By encompassing historical and forthcoming contexts, the

BiGRU can apprehend richer temporal structures and interrelations, thereby enhancing activity recognition performance. Furthermore, the bidirectional processing aids the model in effectively managing intricate and diverse-length sequences, which are prevalent in sensor-based HAR tasks.

### CBAM block

The CBAM proposal was put forth in *Woo et al. (2018)* as an attention mechanism to improve performance by amplifying significant channels and crucial sections of intermediate features. CBAM comprises two sub-modules: the channel and spatial attention modules, as depicted in Fig. 6. These modules are intended for application post-convolutional layers, as their names indicate. CBAM capitalizes on features' spatial and cross-channel connections by progressively integrating channel and spatial attention. To be more specific, it accentuates beneficial channels and reinforces informative local regions.

Figure 6 demonstrates the CBAM attention mechanism, incorporating channel and spatial attention modules. The initial passage of data involves the channel attention module (CAM), illustrated in Fig. 7. CAM employs adaptive learning to assess the importance of each channel, facilitating more precise selection and utilization of information from diverse channels. Subsequently, the data undergoes processing in the spatial attention module, depicted in Fig. 8. This module enhances the network's focus on various spatial positions, enabling a deeper understanding and effectively exploiting significant characteristics in distinct locations. These modules can augment the network's ability to perceive and withstand challenges (*Agac & Durmaz Incel, 2023*).

Figure 7 illustrates the structure of the CAM module. Within the CAM module, both maximum pooling, enhancing sharpness, and average pooling, providing a smoothing effect, are applied in the spatial dimension of the input feature. Subsequently, the input feature is subjected to a multilayer perceptron (MLP) transformation utilizing a reduction ratio ($r$), and eventually, a sigmoid activation function is used. The reduction ratio is crucial in determining the extent of dimensionality reduction. It helps maintain a balance between computational effectiveness and attention accuracy by using a shared MLP component in the channel attention process. A lower reduction ratio can amplify the channel attention mechanism's information expression but escalates computational complexity. Conversely, a higher reduction ratio trims computational complexity but may constrain the channel attention mechanism's expressive capability. Achieving an optimal trade-off between attention performance and computing efficiency necessitates fine-tuning the reduction ratio based on the specific application.

The spatial attention module (SAM) system consists of three consecutive steps, as illustrated in Fig. 8. Initially, two tensors are formed by implementing maximum and average pooling operations on the channels of the input feature $F$. The two tensors are merged and inputted into the convolutional layer with a defined kernel size to generate a feature map with a single channel. Subsequently, the output undergoes processing in the sigmoid activation layer to obtain the final spatial attention mask. Finally, each feature map in $X$ undergoes element-wise multiplication with the constructed spatial attention mask.
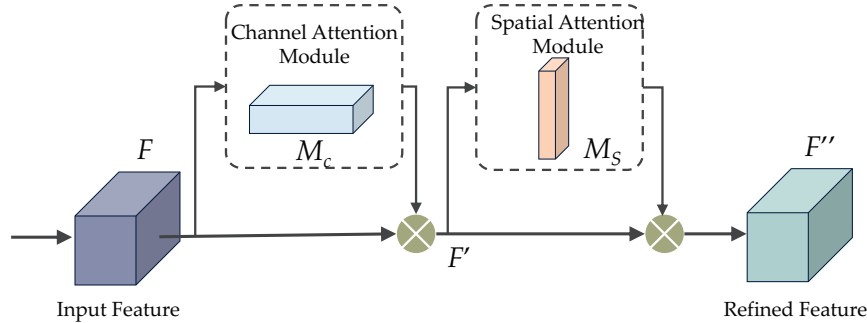

**Figure 6** The convolutional block attention module (CBAM).

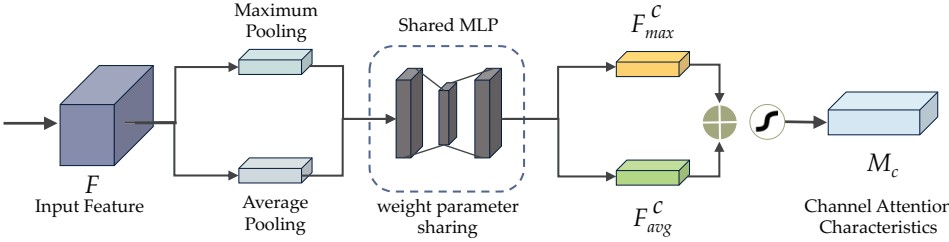

**Figure 7** The channel attention module (CAM) of CBAM.

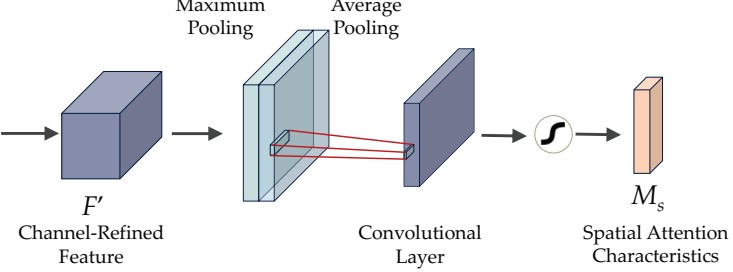

**Figure 8** The spatial attention module (SAM) of CBAM.

## Training and hyperparameters

To ensure the effectiveness of the CNN-BiGRU-CBAM model, having ample and diverse training data is crucial, along with precise adjustment of architectural design parameters known as hyperparameters. These hyperparameters encompass various factors, such as the number of iterations, learning rates, batch size, and activation functions, among others. To ensure the robust and dependable performance of the model, we adopted a conventional approach involving dividing data into distinct training and holdout validation sets. The training set was utilized for hyperparameter optimization, while the holdout validation set was reserved for impartial comparative testing.

By engaging in a systematic process of experimentation and refinement, we identified the most effective hyperparameter configurations that resulted in the highest possible accuracy for the model. The parameters consist of a batch size of 128, a $1 \times 10^{-3}$ learning rate, and an epoch count of 200. Furthermore, we have included an adaptive learning rate technique that decreases the learning rate by 25% if no progress is detected for ten consecutive epochs. This strategy facilitates the model's convergence by enhancing its efficiency and preventing it from being trapped in unsatisfactory alternatives.

We used data shuffling before each training period to increase the model's resilience. This strategy facilitates the model's ability to acquire knowledge from diverse data and mitigates the likelihood of overfitting. For model optimization, we used the Adam optimizer, which dynamically modifies the learning rates of each parameter by taking into account their past gradients. The evaluation of the model's performance was conducted by using cross-entropy loss, a metric that measures the discrepancy between the anticipated probability of the classes and the actual probabilities.

Table 1 comprehensively summarizes the hyperparameters utilized in our custom CNN-BiGRU-CBAM framework. Within the convolutional block are three 1D-convolutional layers, each followed by batch normalization and max pooling steps. These convolutional layers employ a kernel size of 3, a stride of 1, and various filters (256, 128, and 64) to capture information across different scales. The Swish activation function is applied within these convolutional layers to introduce non-linearity and boost the model's expressive capabilities.

For the BiGRU block, 128 hidden units are utilized in both forward and backward directions to capture temporal relationships within the sensor data. To address overfitting, a dropout layer with a dropout rate of 0.25 is applied to the BiGRU outputs. Following the BiGRU block, the CBAM block is appended to enhance the model's ability to focus on the most pertinent features. Ultimately, global average pooling and a dense layer with Softmax activation are employed to generate the final activity predictions.

## Cross validation

To measure the CNN-BiGRU-CBAM model's efficacy, we employed the $k$-fold cross-validation ($k$-CV) method (*Wong, 2015*). This technique involves partitioning the dataset into $k$ subsets of similar size, each distinct and non-overlapping, as shown in Fig. 9. Following this partitioning, one subset is designated as the validation set, while the remaining $k - 1$ subsets are utilized for model training. The overall performance is then determined by averaging performance metrics such as accuracy, precision, recall, and F1-score across all $k$ folds (*Bragança et al., 2022*).

It is worth noting that the $k$-CV procedure can be computationally intensive, especially with large datasets or high $k$ values. In our study, we meticulously balanced computational efficiency and accurate performance estimation by opting for a 5-fold cross-validation ($k$ = 5).

**Table 1  The summary of hyperparameters of the CNN-BiGRU-CBAM used in this work.**

| Stage | Hyperparameters | | Values |
|---|---|---|---|
| Architecture | Convolutional Block | | |
| | 1D-Convolution | Kernel Size | 3 |
| | | Stride | 1 |
| | | Filters | 256 |
| | Activation | | Swish |
| | Batch Normalization | | – |
| | Max Pooling | | 2 |
| | 1D-Convolution | Kernel Size | 3 |
| | | Stride | 1 |
| | | Filters | 128 |
| | Activation | | Swish |
| | Batch Normalization | | – |
| | Max Pooling | | 2 |
| | 1D-Convolution | Kernel Size | 3 |
| | | Stride | 1 |
| | | Filters | 64 |
| | Activation | | Swish |
| | Batch Normalization | | – |
| | Max Pooling | | 2 |
| | BiGRU Block | | |
| | BiGRU Unit | | 128 |
| | Dropout | | 0.25 |
| | CBAM Block | | |
| | CBAM Layer | | – |
| | Global Average Pooling | | – |
| | Dense | | Number of activity classes |
| Training | Loss Function | | Cross-entropy |
| | Optimizer | | Adam |
| | Batch Size | | 128 |
| | Number of Epochs | | 200 |

# EXPERIMENTS AND RESEARCH FINDINGS

In this section, we detail the experiments conducted to pinpoint the most efficient CNN models for recognizing athletic activities from sensor data. Our focus in this study centered on the UCI-DSA benchmark dataset, which is extensively utilized in recognition applications of sport activity. The evaluation of deep learning models in these applications was gauged through commonly recognized metrics, namely accuracy and F1-score.

## Experiment setting

In this research, Google Colab Pro+ with a graphics processing unit module, namely Tesla V100-SXM2-16GB, was employed to accelerate the training of deep learning models. The implementation of CNN-BiGRU-CBAM and other foundational deep learning models

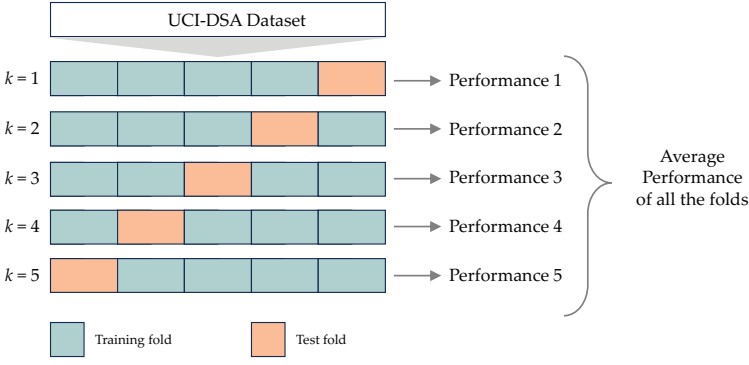

**Figure 9** The 5-fold cross validation.

utilized a Python framework with TensorFlow and CUDA backends. The investigations primarily revolved around the Python packages outlined below:

- Numpy and Pandas were used for data management throughout sensor data retrieval, processing, and analysis.
- Matplotlib and Seaborn were used to create visual representations and display the findings of data analysis and model assessment.
- Scikit-learn was used as a module for collecting and generating data in research.
- TensorFlow was used to generate and train deep learning models.

Various examinations were conducted on the UCI-DSA dataset to determine the most efficient approach. Employing a 5-fold cross-validation strategy, the investigations covered three distinct scenarios involving both sports and everyday activities from the UCI-DSA dataset, as depicted below:

- Scenario I: Exclusively using activities connected to sports (SPT).
- Scenario II: Solely relying on activities that are part of our everyday lives (ADL).
- Scenario III: Incorporating both sport-related and daily living activities (ALL).

## Experimental results

In this part, we detail the evaluation tests carried out, the recognition outcomes achieved by the proposed CNN-BiGRU-CBAM model, and the performance of other comparative deep learning approaches –CNN, long short-term memory (LSTM), bidirectional LSTM (BiLSTM), GRU and BiGRU. Training and testing of the models involved segmented windows extracted from three distinct activity scenarios derived from the publicly available UCI-DSA dataset, containing data from wearable sensors. Scenario I exclusively encompassed sports movements, Scenario II comprised solely sensor data from regular walking, while Scenario III encompassed sports and daily activities. The models underwent assessment for accuracy and F1-score in multi-class classification through 5-fold stratified cross-validation.

A benchmark test was conducted with various configurations of wearable sensors on the torso, arms, and legs to gauge the impact of sensor placement on the body. The

**Table 2   Recognition performance of deep learning models on Scenario I using sensor data of sport activities from different body positions.**

| Model | Recognition effectiveness | | | | | | | | | |
| | Torso | | Right arm | | Left arm | | Right leg | | Left leg | |
| | Accuracy | F1-score | Accuracy | F1-score | Accuracy | F1-score | Accuracy | F1-score | Accuracy | F1-score |
|---|---|---|---|---|---|---|---|---|---|---|
| CNN | 99.46 | 99.46 | 99.17 | 99.17 | 99.15 | 99.14 | 99.85 | 99.85 | 99.71 | 99.71 |
| LSTM | 99.69 | 99.69 | 98.85 | 98.86 | 98.63 | 98.61 | 99.92 | 99.92 | 99.96 | 99.96 |
| BiLSTM | 99.73 | 99.73 | 99.19 | 99.19 | 99.23 | 99.23 | 99.92 | 99.92 | 100.00 | 100.00 |
| GRU | 99.71 | 99.71 | 99.29 | 99.29 | 99.25 | 99.25 | 99.98 | 99.98 | 99.88 | 99.87 |
| BiGRU | 99.67 | 99.67 | 99.38 | 99.37 | 99.40 | 99.39 | 99.94 | 99.94 | 99.96 | 99.96 |
| CNN–BiGRU-CBAM | 99.94 | 99.94 | 99.81 | 99.81 | 99.92 | 99.92 | 99.98 | 99.98 | 100.00 | 100.00 |

**Table 3   Recognition performance of deep learning models on Scenario II using sensor data of daily activities from different body positions.**

| Model | Recognition effectiveness | | | | | | | | | |
| | Torso | | Right arm | | Left arm | | Right leg | | Left leg | |
| | Accuracy | F1-score | Accuracy | F1-score | Accuracy | F1-score | Accuracy | F1-score | Accuracy | F1-score |
|---|---|---|---|---|---|---|---|---|---|---|
| CNN | 97.36 | 97.32 | 97.34 | 97.31 | 97.66 | 97.65 | 97.48 | 97.43 | 97.64 | 97.60 |
| LSTM | 97.18 | 97.13 | 96.34 | 96.29 | 96.41 | 96.36 | 97.34 | 97.28 | 97.62 | 97.57 |
| BiLSTM | 97.87 | 97.84 | 97.36 | 97.29 | 97.41 | 97.39 | 98.01 | 97.99 | 97.80 | 97.78 |
| GRU | 97.13 | 98.11 | 7.59 | 97.57 | 97.85 | 97.82 | 98.08 | 98.07 | 98.08 | 98.06 |
| BiGRU | 98.31 | 98.30 | 97.25 | 97.21 | 98.36 | 98.34 | 98.52 | 98.51 | 98.36 | 98.35 |
| CNN–BiGRU-CBAM | 98.36 | 98.36 | 97.99 | 97.96 | 98.59 | 98.58 | 98.01 | 97.99 | 98.38 | 98.36 |

**Table 4   Recognition performance of deep learning models on Scenario III using sensor data of sport and daily activities from different body positions.**

| Model | Recognition effectiveness | | | | | | | | | |
| | Torso | | Right arm | | Left arm | | Right leg | | Left leg | |
| | Accuracy | F1-score | Accuracy | F1-score | Accuracy | F1-score | Accuracy | F1-score | Accuracy | F1-score |
|---|---|---|---|---|---|---|---|---|---|---|
| CNN | 97.35 | 97.27 | 97.28 | 97.24 | 97.77 | 97.77 | 97.91 | 97.89 | 98.26 | 98.23 |
| LSTM | 98.32 | 98.31 | 97.41 | 97.37 | 97.74 | 97.71 | 98.55 | 98.54 | 98.73 | 98.71 |
| BiLSTM | 98.61 | 98.61 | 97.76 | 97.74 | 97.29 | 97.18 | 98.87 | 98.86 | 99.07 | 99.06 |
| GRU | 98.53 | 98.52 | 97.88 | 97.84 | 98.25 | 98.24 | 98.84 | 98.83 | 99.05 | 99.03 |
| BiGRU | 98.62 | 98.61 | 98.39 | 98.38 | 98.67 | 98.67 | 98.90 | 98.89 | 98.49 | 98.48 |
| CNN–BiGRU-CBAM | 98.62 | 98.62 | 98.75 | 98.74 | 98.98 | 98.98 | 98.91 | 98.91 | 99.10 | 99.10 |

experimental results, revealing the model's performance, were presented in Tables 2, 3 and 4.

As per the findings in Table 2, the CNN-BiGRU-CBAM model demonstrates the highest accuracy in classification and F1-scores across all five body locations: torso, right arm, left arm, right leg, and left leg. The model consistently achieves accuracy exceeding 99.8% when utilizing sensor data from any body part. This highlights the model's capability to accurately recognize diverse athletic movements, irrespective of variations in positional movement patterns. In contrast, baseline deep learning models like CNN and LSTM

typically achieve accuracy levels ranging between 98% and 99%, with variability based on sensor positioning. Notably, their performance experiences a significant drop when relying solely on data from either the left or right arm.

Moreover, there is a slight improvement in results for all models when utilizing data from the right leg compared to the left leg. This hints at a subtle asymmetry in body movements during athletic activities, providing more informative motion signals that the models can discern from the right side limbs. In essence, despite potential variations in accuracy tied to sensor placement, the outcomes affirm that the CNN-BiGRU-CBAM model can effectively classify various sports exercises using simplified wearable sensors placed on any monitored body part without significant information loss. This adaptability proves beneficial for practical sporting applications requiring precise sensor positioning.

Table 3 displays the outcomes of recognizing everyday activities (Scenario II) using diverse placements of on-body sensors. The CNN-BiGRU-CBAM method achieves the highest accuracy across all five observed body regions. However, unlike in sports activities, variations emerge depending on sensor positioning. To be more specific, sensors positioned on the torso yield the most valuable signals for everyday activity recognition, registering an accuracy of 98.36% with the CNN-BiGRU-CBAM model. Following closely behind is the data from the left arm. Conversely, leg measurements provide less reliable information, resulting in a roughly 2% decrease in accuracy. Additionally, when comparing signals from the arms, it has been demonstrated that left-arm sensors consistently exhibit a 2–5% higher accuracy compared to their right-arm counterparts across various everyday tasks, irrespective of the model used. This observed laterality pattern reverses when contrasted with the movements associated with sports activities.

The CNN-BiGRU-CBAM model accurately identifies a combined dataset of sports and everyday life activities using wearable motion data, as shown in Table 4. This superior performance extends across all five monitored body postures. However, the most effective sensor placement varies compared to the studies focused on specific categories.

In Scenario III, where a diverse range of actions is considered, the right arm sensors prove particularly valuable, contributing to an impressive accuracy of 98.75% with the CNN-BiGRU-CBAM model. This surpasses the previously observed torso placement reliability for isolated sports and regular activities. Arm-worn sensors consistently demonstrate robust identification capabilities, with the left arm achieving an exceptional accuracy of 98.98% using the CNN-BiGRU-CBAM method. Therefore, upper limb dynamics prove most suitable for capturing the myriad movements in this broad activity set, ranging from basic periodic motions like walking to sophisticated multi-state sports workouts.

Conversely, data from the analysis of leg movements yields comparatively fewer significant insights for categorizing this diverse array of motions. Despite this, the proposed technique consistently attains an accuracy rate exceeding 98.9% across all body postures, establishing its reliable capability for recognizing various activities.

### Assessing the model's learning capacity and overfitting resilience

To grasp the learning patterns of the CNN-BiGRU-CBAM model, we analyzed the validation accuracy during various training phases across three specific scenarios: (1)

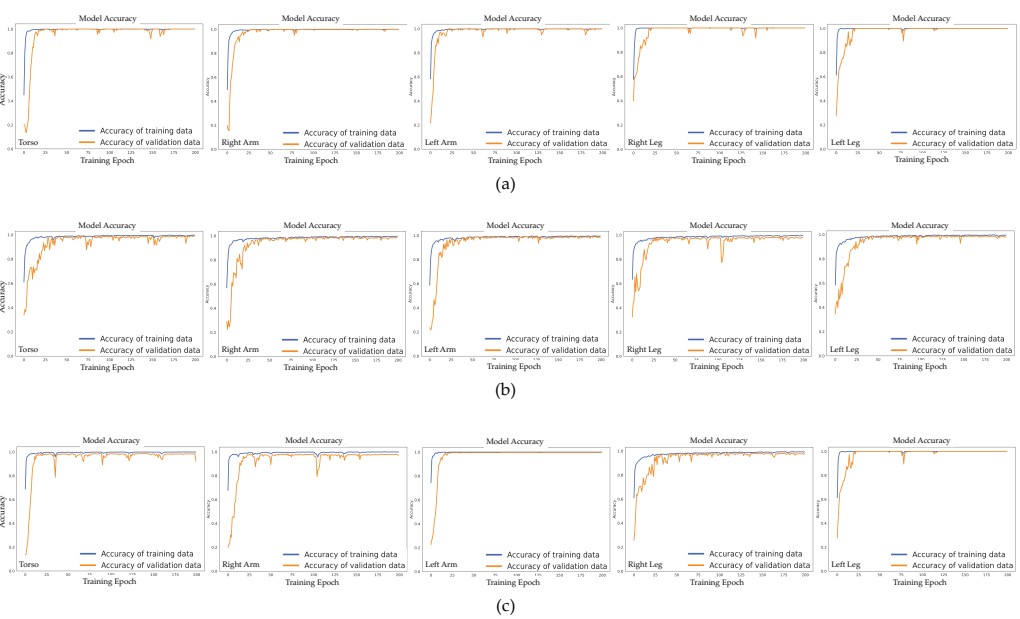

**Figure 10** Validation accuracy of the proposed CNN-BiGRU-CBAM model across training epochs for different scenarios and body positions: (A) Scenario I: using only sensor data from sport activities, (B) Scenario II: using only sensor data from activities of daily living, and (C) Scenario III: using sensor data from both sport activities and activities of daily living.

employing sensor data exclusively from sports activities, (2) using sensor data exclusively from daily life activities, and (3) leveraging sensor data from both sports activities and daily life activities. This investigation helps to understand the model's capacity to adapt to new data and could detect issues such as overfitting or underfitting.

Figure 10 visually depicts the relationship between epochs and validation accuracy within the CNN-BiGRU-CBAM model. This model incorporates sensor data from various body regions, including the torso, right arm, left arm, right leg, and left leg, across different contexts. Across all three scenarios examined, our model consistently improves its validation accuracy as epochs progress, reaching a notable performance level around the 50th epoch. The validation accuracy stabilizes after that, suggesting the model has reached a dependable solution. Figure 10 also illustrates the model's learning trends and generalization capabilities, showcasing a steady increase in validation accuracy over epochs and resilience against overfitting across different scenarios and body positions.

In the first scenario (Fig. 10A), the CNN-BiGRU-CBAM model achieves peak validation accuracy when utilizing sensor data from the left leg, closely followed by the right leg and torso data. While the accuracy slightly diminishes with data from both arms, it remains substantial. This observation suggests that movements of the legs are crucial for discerning athletic activities.

In scenario II (Fig. 10B), the model's performance remains relatively steady across different body postures, with slightly higher validation accuracies observed for data from

the left arm and right leg. This indicates that sensor data from distinct body orientations have similar impacts on identifying daily life activities.

In Fig. 10C, Scenario III demonstrates that the CNN-BiGRU-CBAM model achieves optimal validation accuracy when using data from the left leg, followed by the right arm and left arm data. Although the accuracy diminishes somewhat with data from the torso and right leg, it remains robust. These findings suggest that combining motions of the legs and arms yields distinctive characteristics for identifying both sports and everyday activities.

The charts depicting epoch *versus* validation accuracy for all three scenarios reveal that the CNN-BiGRU-CBAM model exhibits no signs of overfitting. This is evidenced by the stable validation accuracy even after numerous epochs. This observation indicates that the model has acquired resilient and significant features from the sensor data, enabling it to maintain high-performance levels on unseen data.

## Comparison results with state-of-the-art models

To demonstrate the exceptional performance of our CNN-BiGRU-CBAM model, we thoroughly compared it with many cutting-edge models in the field of individual activity identification utilizing data from wearable sensors. The selections for benchmark models comprise:

- GoogleLeNet (*Szegedy et al., 2015*), a CNN model, incorporates the inception module, facilitating rapid computation and improved performance by aggregating multi-scale information.
- ResNeXt (*Xie et al., 2017*), an advanced iteration of the ResNet architecture, integrates the inception module with a residual framework, enabling the network to encompass a broader spectrum of intricate and diverse characteristics.
- Multi-STMT (*Zhang & Xu, 2024*) is a sophisticated network composition comprising a CNN component, a BiGRU component, and an attention mechanism. Its objective is to capture spatial and temporal relationships within sensor input data.

We evaluated the efficacy of our proposed model compared to conventional ones using the UCI-DSA dataset, assessing two key metrics: accuracy and F1-score. The results, presented in Table 5, demonstrate that our CNN-BiGRU-CBAM model outperforms state-of-the-art methods, achieving an average accuracy of 99.51% and an F1-score of 99.51%.

Among all the models employed for comparison, Multi-STMT exhibits the closest performance to our proposed model, boasting an impressive accuracy rate of 99.39% and an outstanding F1-score of 99.49%. This effectiveness can be attributed to the multi-tiered structure of the system, which integrates CNN and BiGRU modules alongside an attention mechanism. Such architecture enables accurate capture of spatial and temporal relationships within the sensor data. However, our CNN-BiGRU-CBAM model outperforms Multi-STMT by a slight margin, underscoring the efficacy of our meticulously designed framework. Our approach utilizes convolutional blocks to extract spatial features, BiGRU for temporal context modeling, and CBAM for feature refinement

**Table 5  Comparative analysis of the proposed CNN-BiGRU-CBAM model against state-of-the-art models using wearable sensor data on the UCI-DSA dataset.**

| Model | Recognition Performance | |
|---|---|---|
| | Accuracy | F1-score |
| GoogleLeNet (*Szegedy et al., 2015*) | 96.64% | 96.36% |
| ResNeXt (*Xie et al., 2017*) | 98.81% | 98.82% |
| Multi-STMT (*Zhang & Xu, 2024*) | 99.39% | 99.49% |
| The proposed CNN-ResBiGRU-CBAM | 99.51% | 99.51% |

through attention. This amalgamation of techniques empowers our model to adeptly discern and utilize discriminative patterns within the sensor data, thereby augmenting identification accuracy.

The UCI-DSA dataset employed in this comparative study is esteemed for its extensive array of characteristics, which are attributed to the plethora of sensors utilized during data collection. This abundance of features facilitates efficient feature extraction by the models, contributing to the outstanding classification accuracy observed across most approaches applied to this dataset.

## DISCUSSION

In this part, we thoroughly examine the research observations discussed in the previous section.

### Impact of different types of activities

Building upon the experimental results, we explored how different types of activities impact the classification performance of the proposed CNN-BiGRU-CBAM model in distinguishing between sports and everyday activities. The outcomes from Tables 2 to 4 were analyzed and visually represented through bar graphs in Fig. 11.

Presented in Fig. 11 is a comparison of the CNN-BiGRU-CBAM model's accuracy in multi-class classification across three activity scenarios. The model relies on data from body-worn sensors positioned in diverse body postures. Specifically, it achieves an accuracy of 99.94% for sports movements, showcasing its highly reliable capability in recognizing a diverse array of intricate exercise routines. This performance slightly diminishes to 98.36% for everyday ambulation, as these routine activities entail less motion complexity that may challenge deep networks.

Remarkably, when considering the combination of sports and everyday activities, the accuracy remains consistent with the scenario focused solely on sports, maintaining a precision of 98.62%. This underscores the model's adeptness in accurately categorizing complex real-life situations encompassing intricate sports activities and routine daily movements.

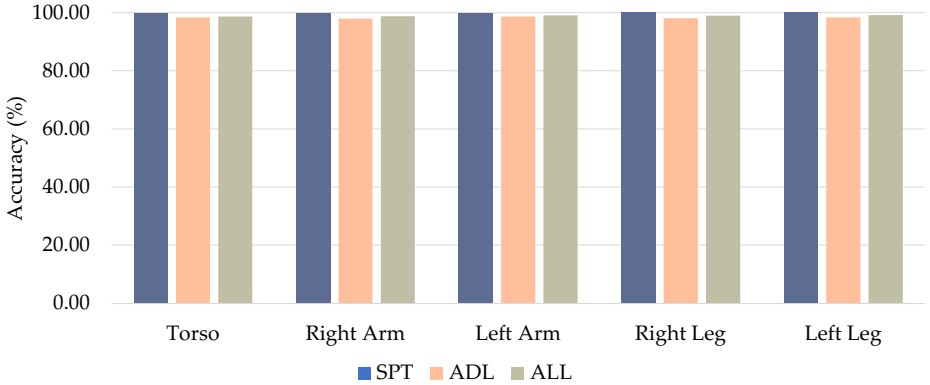

**Figure 11** **Comparison results of the proposed CNN-BiGRU-CBAM using sensor data from different activity types.**

| Scenario | Torso | Right Arm | Left Arm | Right Leg | Left Leg |
|---|---|---|---|---|---|
| SPT | 99.94 | 99.81 | 99.92 | 99.98 | 100.00 |
| ADL | 98.36 | 97.99 | 98.59 | 98.01 | 98.38 |
| ALL | 98.62 | 98.75 | 98.98 | 98.91 | 99.10 |

## Impact of different body positions for sport and daily activity recognition

The visual comparison illustrates the CNN-BiGRU-CBAM model's accuracy in categorizing activities. This model relies on sensor data from five wearable positions: torso, right arm, left arm, right leg, and left leg. In the context of routine tasks, the torso emerges as the optimal location, achieving a high 98.36% accuracy. This aligns with expectations, given that typical walking primarily involves upper body movements.

However, when it comes to sports movements, the accuracy of torso-based sensors drops to 99.94%, somewhat lower than the 99.98–100% achieved by leg-worn sensors. This implies that the enhanced mobility and agility in sports activities make legs more effective in capturing intricate routines. Combining both activity categories, placing the right arm strikes an optimal balance, maximizing information availability for identifying diverse activities. This emphasizes the practicality of wrist-worn devices commonly found in consumer wearables.

In brief, torso motion is beneficial for sensing everyday activities, whereas limb placement proves more suitable for sports analytics. Arm signals determine generalized positioning, which is relevant for accurate deep activity detection across various scenarios.

## Ablation studies

In the realm of neural networks (*Montaha et al., 2021*), the concept of ablation study has gained prominence as a means to evaluate a model's effectiveness by investigating how altering specific elements affects its performance (*de Vente et al., 2020*). Consequently, we scrutinize the effects of ablation on our proposed model through three distinct research scenarios. In these instances, we adjust various blocks and layers to evaluate their impact on the proposed design (*Meyes et al., 2019*). As a result, upon completing all research scenarios,

**Table 6   Impact of the convolution block.**

| Model | Recognition Performance | | |
|---|---|---|---|
| | **Accuracy** | **Loss** | **F1-score** |
| CNN-BiGRU-CBAM without Convolutional Block | 98.65%(±0.23%) | 0.05(±0.02) | 98.65%(±0.23%) |
| The proposed CNN-ResBiGRU-CBAM | 99.51%(±0.33%) | 0.02(±0.01) | 99.51%(±0.33%) |

we aim to identify the most optimal configuration of our proposed CNN-BiGRU-CBAM model, ultimately leading to maximal recognition performance.

### Impact of the convolution block

To explore the impact of the convolutional block on model performance, we conducted an ablation study using the UCI-DSA dataset. This involved creating a baseline model by excluding the convolutional block from the original CNN-BiGRU-CBAM architecture while retaining the BiGRU and CBAM blocks. The baseline model directly feeds raw sensor data into the BiGRU block without any spatial feature extraction.

Table 6 presents the findings of the ablation investigation. The model without the convolutional blocks exhibits inferior recognition performance compared to the complete CNN-BiGRU-CBAM model. Removing the convolutional blocks decreases the F1-score from 99.51% to 98.65%, underscoring the importance of spatial feature extraction for accurate activity identification.

### Impact of the BiGRU block

The BiGRU component within our CNN-BiGRU-CBAM framework plays a crucial role in capturing temporal connections and context from the features extracted by the CNN layers. GRUs are a specific type of RNN renowned for processing sequential data by maintaining internal states that retain and recall information from previous time steps.

Our model employs a BiGRU, comprising two GRU layers that process the input sequence in both forward and backward directions. The forward GRU sequentially analyzes the input sequence from the initial to the final time step, while the backward GRU examines the sequence in reverse, from the final to the initial time step. This bidirectional approach allows the BiGRU to capture context from past and future time steps, providing a comprehensive understanding of temporal relationships between the extracted features.

The BiGRU component receives the CNN layers' output as input, representing spatial characteristics derived from sensor data. The input sequence undergoes separate processing by the forward and backward GRU layers, with their outputs combined at each time step to form the final BiGRU output. This output encapsulates sequential connections and contextual information from the extracted features, facilitating a deeper comprehension of the relationships between different actions and enhancing overall identification accuracy.

To evaluate the BiGRU block's effectiveness in capturing temporal relationships, we conducted an ablation experiment on our proposed CNN-BiGRU-CBAM model. The baseline architecture for this experiment was a modified version of CNN-BiGRU-CBAM

**Table 7  Impact of the BiGRU block.**

| Model | Recognition performance | | |
|---|---|---|---|
| | **Accuracy** | **Loss** | **F1-score** |
| CNN-BiGRU-CBAM without BiGRU Block | 98.24%(±0.53%) | 0.05(±0.01) | 98.24%(±0.54%) |
| The proposed CNN-ResBiGRU-CBAM | 99.51%(±0.33%) | 0.02(±0.01) | 99.51%(±0.33%) |

lacking a BiGRU block. The results in Table 7 confidently indicate that the CNN-BiGRU-CBAM model, incorporating the BiGRU block, outperforms the baseline model without it. Specifically, it achieves an increase of approximately 1.27% in F1-score on the UCI-DSA dataset.

The BiGRU block's bidirectional processing capability is advantageous for activity recognition tasks as it integrates information from past and future time steps, which is crucial for distinguishing between various activities. By capturing relevant context, the BiGRU block enhances the model's ability to accurately identify and categorize different sports and daily activities using sensor data.

### Impact of the CBAM block

The CBAM block, a novel addition to our proposed CNN-BiGRU-CBAM model, is designed to enhance the model's ability to focus on crucial features for activity detection. This is achieved by implementing attention mechanisms in both the channel and spatial dimensions, a unique approach in the field of computer vision and machine learning. This block consists of two consecutive submodules: the channel and spatial attention modules. The former exploits inter-channel interactions to generate a channel attention map, emphasizing informative channels, while the latter utilizes inter-spatial interactions to develop a spatial attention map, highlighting relevant spatial locations.

Positioned after the BiGRU block in our model, the CBAM block refines spatial–temporal feature representations before reaching the final classification layer. By employing attention mechanisms, the CBAM block enables the model to prioritize essential areas in feature maps while filtering out noise and irrelevant data. This attention-based feature refinement is believed to enhance the model's ability to detect activities effectively across various body postures, a significant advancement in the field of activity detection.

To evaluate the impact of the CBAM block, we conducted an ablation study, removing the CBAM block from the CNN-BiGRU-CBAM model while retaining the convolutional and BiGRU modules. The performance of the ablated model was assessed on all three benchmark datasets and compared to the complete CNN-BiGRU-CBAM model.

The results of the ablation study, presented in Table 8, demonstrate that the CNN-BiGRU-CBAM model incorporating the CBAM block achieves superior recognition performance compared to the ablated model without the CBAM block. Integration of the CBAM block leads to a 0.36% improvement in the F1-score for the UCI-DSA dataset. This enhancement can be attributed to the attention mechanism's ability to highlight important features and reduce interference, thereby enhancing the model's ability to discriminate between different elements.

**Table 8  Impact of the CBAM block.**

| Model | Recognition Performance | | |
|---|---|---|---|
| | Accuracy | Loss | F1-score |
| CNN-BiGRU-CBAM without CBAM Block | 99.14%(±0.28%) | 0.03(±0.02) | 99.14%(±0.28%) |
| The proposed CNN-ResBiGRU-CBAM | 99.51%(±0.33%) | 0.02(±0.01) | 99.51%(±0.33%) |

**Table 9  Comparison of the number of trainable parameters, memory consumption, and mean prediction times for baseline deep learning models (CNN, LSTM, BiLSTM, GRU, and BiGRU) and the proposed CNN-BiGRU-CBAM model using the benchmark UCI-DSA dataset.**

| Model | Parameters | Memory (bytes) | Mean prediction time (ms) |
|---|---|---|---|
| CNN | 509,011 | 6,147,880 | 0.190 |
| LSTM | 108,051 | 1,339,032 | 0.270 |
| BiLSTM | 279,315 | 3,412,376 | 0.425 |
| GRU | 86,163 | 1,076,376 | 0.278 |
| BiGRU | 169,747 | 2,091,160 | 0.416 |
| CNN-BiGRU-CBAM | 312,698 | 3,914,472 | 0.425 |

In summary, the CNN-BiGRU-CBAM model employs convolutional modules, a BiGRU component, and a CBAM module to effectively extract the most relevant features from the data captured by wearable sensors.

## Complexity analysis

To thoroughly explore the algorithm's intricacies, we conducted an extensive examination of the proposed CNN-BiGRU-CBAM model. This analysis compared it with baseline deep learning models such as CNN, LSTM, BiLSTM, GRU, and BiGRU. We adopted the HAR evaluation methodology outlined by *Angerbauer et al. (2021)* for this study. The complexity was assessed by measuring the models' memory usage, average prediction time, and the number of trainable parameters. The benchmark UCI-DSA dataset utilized in this investigation was employed to assess all models.

### *Memory consumption*

During the inference process on test dataset batches, we evaluated the memory usage of the CNN-BiGRU-CBAM architecture compared to the baseline CNN, LSTM, BiLSTM, GRU, and BiGRU architectures using CUDA profiling tools. The results presented in Table 9 reveal that our proposed model requires an average of 3,914,472 bytes of memory. This memory stores encoder feature maps, temporal hidden states, classifier weights, and outputs. In contrast, the basic CNN consumes an average of 6,147,880 bytes when processing shorter feature sequences. Conversely, the standalone BiGRU architecture occupies an average of 2,091,160 bytes without employing convolutions to represent spatial hierarchies. The CNN-BiGRU-CBAM model strikes a balance by enhancing accuracy without substantially increasing memory usage.

### Prediction time

To evaluate efficiency, we analyze the complexity of the models based on their mean prediction time. This metric is determined by feeding a set of samples from the test data into the HAR models and calculating the average duration required to process one window. The results of this analysis are presented in Table 9, showcasing the average prediction time in milliseconds for the deep learning models operating on the UCI-DSA dataset. BiGRU exhibits the most prolonged prediction latency among these models, with a duration of 0.416 ms. It is worth noting that BiGRUs typically require a significantly longer time during training than CNN-based models, including our proposed CNN-BiGRU-CBAM. This is because convolutions can be executed concurrently, only needing a few adjacent values to compute the output of a single kernel. However, with BiGRU, a substantial portion of computations must be performed sequentially, as outputs depend on prior outputs. The mean prediction time of the CNN-BiGRU-CBAM model is 0.425 ms, comparable to that of the BiGRU and BiLSTM models.

### Trainable parameters

In addition to memory usage and average prediction time, we consider the number of trainable parameters to indicate model complexity. These parameters represent the weights that are adjusted during the model training process, and a higher count suggests the model's capacity to convey more intricate data. However, if the data lacks complexity, it also increases the risk of overfitting.

The count of trainable parameters for the deep learning models (CNN, LSTM, BiLSTM, GRU, BiGRU, and the proposed CNN-BiGRU-CBAM) utilized in this study is meticulously presented in Table 9. These values were obtained from the model summary using the benchmark UCI-DSA dataset, ensuring the accuracy and reliability of our findings. The results align with the intuitive understanding of the complexity differences among these models. Among the fundamental deep learning models, the GRU is the simplest, with 86,163 parameters, while the CNN is the most complex, with 509,011 parameters. The proposed CNN-BiGRU-CBAM model contains a total of 312,698 trainable parameters. This figure is notably higher than those of other deep learning models but remains lower than that of the CNN model.

## CONCLUSION AND FUTURE WORKS

This research delved into applying advanced learning approaches to accurately recognize diverse sports and fitness activities using information gathered from wearable sensors. An essential contribution lies in introducing an integrated CNN-BiGRU-CBAM model, amalgamating convolutional, recurrent, and attention-based neural network structures. The primary goal is to boost the precision and dependability of detecting sports and everyday activities. The performance of this hybrid model was assessed using the UCI-DSA benchmark dataset, showcasing its capability to categorize a broad spectrum of intricate workout movements precisely. Notably, the proposed technique outperforms comparable deep learning benchmarks like CNN, LSTM, BiLSTM, GRU, and BiGRU. Rigorous

evaluations validate its consistent and accurate recognition across numerous classes in both sports and everyday activities, relying solely on sensor data.

Ongoing research efforts are crucial to bolster the robustness, interpretability, and accessibility of models for practical deployment. Nonetheless, intelligent activity identification holds substantial potential to revolutionize training, competition, and rehabilitation in sports. Potential future directions also include:

- Deploying models on portable edge devices for minimal analysis delays,
- Integrating sensor data with video streams and
- Exploring attention methods to enhance model transparency and comprehensibility.

### Funding

This research was funded by the Thailand Science Research and Innovation Fund; University of Phayao (Grant No. FF67-UoE-Sakorn); and King Mongkut's University of Technology North Bangkok with Contract no. KMUTNB-67-KNOW-12. The funders had no role in study design, data collection and analysis, decision to publish, or preparation of the manuscript.

### Grant Disclosures

The following grant information was disclosed by the authors:
Thailand Science Research and Innovation Fund.
University of Phayao: FF67-UoE-Sakorn.
King Mongkut's University of Technology North Bangkok: KMUTNB-67-KNOW-12.

### Competing Interests

The authors declare there are no competing interests.

### Author Contributions

- Sakorn Mekruksavanich conceived and designed the experiments, performed the experiments, performed the computation work, authored or reviewed drafts of the article, and approved the final draft.
- Wikanda Phaphan analyzed the data, prepared figures and/or tables, and approved the final draft.
- Narit Hnoohom performed the computation work, prepared figures and/or tables, and approved the final draft.
- Anuchit Jitpattanakul conceived and designed the experiments, performed the computation work, authored or reviewed drafts of the article, and approved the final draft.

### Data Availability

The data is available at Kerem Altun, Billur Barshan, February 15, 2019, '' Daily and Sports Activities Data Set", IEEE Dataport, doi: https://dx.doi.org/10.21227/at1v-6f84.

The code is available in the Supplementary File.

## Supplemental Information

Supplemental information for this article can be found online at http://dx.doi.org/10.7717/peerj-cs.2100#supplemental-information.

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
