# Peer review of "Recognition of sports and daily activities through deep learning and convolutional block attention"

_PeerJ Computer Science, doi:10.7717/peerj-cs.2100_

## Round 0.1 · original submission · Major Revisions

Dear authors,

You are advised to critically respond to all comments point by point when preparing a new version of the manuscript and while preparing for the rebuttal letter. Please address all the comments/suggestions provided by the reviewers.

Some reviewers have suggested that you cite specific references. You are welcome to add it/them if you believe they are relevant. However, you are not required to include these citations, and if you do not include them, this will not influence my decision.

Kind regards,
PCoelho

**Language Note:** PeerJ staff have identified that the English language needs to be improved. When you prepare your next revision, please either (i) have a colleague who is proficient in English and familiar with the subject matter review your manuscript, or (ii) contact a professional editing service to review your manuscript. PeerJ can provide language editing services - you can contact us at [email protected] for pricing (be sure to provide your manuscript number and title). – PeerJ Staff

Reviewer 1 ·

Basic reporting

This research delved into applying advanced learning approaches to accurately recognize diverse sports and fitness activities using information gathered from wearable sensors. An essential contribution lies
in introducing an integrated CNN-BiGRU-CBAM model, amalgamating convolutional, recurrent, and
attention-based neural network structures. The primary goal is to boost the precision and dependability
of detecting sports and everyday activities. The performance of this hybrid model was assessed using
the UCI-DSA benchmark dataset, showcasing its capability to categorize a broad spectrum of intricate
workout movements precisely. My main concerns are as follows:
1. As the related works and the state-of-the-art are concerned, the survey that comes with the manuscript is, in my opinion, quite scarce; furthermore, enhancements, advantages and potential disadvantages of the proposed frameworks regarding the state-of-the-art are not discusses. I suggest including the follow references into discussion, such as https://doi.org/10.1145/3631355;DOI: 10.1109/TITS.2023.3309600;https://doi.org/10.1145/3568679;https://doi.org/10.1016/j.eswa.2023.122898. A comparison of the approach with similar works from the state-of-the-art, both from the qualitative and quantitative perspective, would be really appreciated.

2. What is your proposed method? It is confused whether the authors have improved the existing algorithms. If so, please clearly indicate the improvement or modifications.The authors need to make a clear explanation of the main research contribution of this manuscript, as well as its state-of-art research methods and research content.
3. The ablation experiments hould be presented as a separate paragraph to highlight that the authors' approach is robust enough.
4. It is necessary to provide a detailed analysis of the algorithm complexity in section the Proposed CNN-BiGRU-CBAM Model.
4. The presented comparison results lack for technical details that can justify the obtained numerical results in section Experimental Results. Practical implementation needed to be elaborated the merit of your proposed framework.

Experimental design

Due to the latest work in this field, the experimental part of this article is relatively weak, with only comparative experiments with different parameters and ablation experiments. However, some effort can be made in theoretical proof.

Validity of the findings

N/A

Additional comments

N/A

Reviewer 2 ·

Basic reporting

Professional language followed. Sufficient Literaure used.

Experimental design

How are the control parameters set?
Methods are not defined with sufficient detail. Architecure in particular.

Validity of the findings

Weak discussion
How to validat the model?
Any comparison with previous models with ablation study or CV will be better

Additional comments

1. Abstract needs revision. Novelty, number of samples tested etc must be fitted in
2, Lit review : Implications from previous findings, gaps identified, how this work addresses those must be spelled
3. What is the datasplit followed?
4. Why the three methods are employed must be justified correctly?
5.

Reviewer 3 ·

Basic reporting

1) Please write the problem statement or research gap ( What you want to do and why) in introduction clearly.
2) Write the literature review group wise. Please make group with similar type of papers and the review those papers analytically.
3) Based on these review write your contributions very specifically at the end of Introduction.

Experimental design

1) How does Bidirectional GRU work in your model? Please explain.

Validity of the findings

1) You need to do more experiments : Please compare your work with some other existing similar works.
2) Show your results of Epochs vs Validation Accuracy.
3) Show your results with K-fold cross validation.
4) Please read and cite: a) An automated daily sports activities and gender recognition method based on novel multikernel local diamond pattern using sensor signals (2020), b) Deep CNN-LSTM with Self-Attention Model for Human Activity Recognition using Wearable Sensors (2022), c) GRU-INC: An Inception-Attention based Approach using GRU for Human Activity Recognition (2023), d) A review of multimodal human activity recognition with special emphasis on classification, applications, challenges and future directions (2021), e) Trends in human activity recognition with focus on machine learning and power requirements (2021), f) Deep CNN-GRU Based Human Activity Recognition with Automatic Feature Extraction Using Wearable and Smartphone Sensors (2023)

Additional comments

None

---

## Round 0.2 · accepted · Accept

Dear authors, we are pleased to verify that you meet the reviewer's valuable feedback to improve your research.

Thank you for considering PeerJ Computer Science and submitting your work.

Reviewer 1 ·

Basic reporting

good

Experimental design

good

Validity of the findings

good

Additional comments

good

Reviewer 2 ·

Basic reporting

No more suggestions

Experimental design

No more suggestions

Validity of the findings

No more suggestions

Additional comments

No more suggestions

Reviewer 3 ·

Basic reporting

I think the authors have addressed all the comments successfully.

Experimental design

I think the authors have addressed all the comments successfully

Validity of the findings

I think the authors have addressed all the comments successfully